# AN INFERENCE-BASED POLICY GRADIENT METHOD FOR LEARNING OPTIONS

## ABSTRACT

In the pursuit of increasingly intelligent learning systems, abstraction plays a vital role in enabling sophisticated decisions to be made in complex environments. The options framework provides formalism for such abstraction over sequences of decisions. However most models require that options be given a priori, presumably specified by hand, which is neither efficient, nor scalable. Indeed, it is preferable to learn options directly from interaction with the environment. Despite several efforts, this remains a difficult problem: many approaches require access to a model of the environmental dynamics, and inferred options are often not interpretable, which limits our ability to explain the system behavior for verification or debugging purposes. In this work we develop a novel policy gradient method for the automatic learning of policies with options. This algorithm uses inference methods to simultaneously improve all of the options available to an agent, and thus can be employed in an off-policy manner, without observing option labels. Experimental results show that the options learned are interpretable. Further, we find that the method presented here is more sample efficient than existing methods, leading to faster and more stable learning of policies with options.

## 1 INTRODUCTION

Recent developments in reinforcement learning (RL) methods have enabled agents to solve problems in increasingly complicated domains (Mnih et al., 2016; Mousavi et al., 2018). However, in order for agents to solve more difficult and realistic environments–potentially involving long sequences of decisions–more sample efficient techniques are needed. One way to improve on existing agents is to leverage abstraction. By reasoning at various levels of abstraction, it is possible to infer, learn and plan much more efficiently. Recent developments have lead to breakthroughs in terms of learning rich representations for perceptual information (Bengio et al., 2013). In RL domains, however, while efficient methods exist to plan and learn when abstraction over sequences of actions is provided *a priori*, it has proven more difficult to learn this type of temporal abstraction from interaction data.

Many frameworks for formalizing temporal abstraction have been proposed; most recent developments build on the *Options* framework (Sutton et al., 1999; Precup, 2000), which offers a flexible parameterization, potentially amenable to learning. The majority of prior work on learning options has centered around the idea of discovering *subgoals* in state space, and constructing a set of options such that each option represents a policy leading to that subgoal (McGovern & Barto, 2001; Menache et al., 2002; Şimşek & Barto, 2009). These methods can lead to useful abstraction, however, they often require access to a model of the environment dynamics, which is not always available, and can be infeasible to learn. Our contributions instead build on the work of Bacon et al. (2017), and exploits a careful parameterization of the policy of the agent, in order to simultaneously learn a set of options, while directly optimizing returns. We relax a few key assumptions of this previous work, including the expectation that only options that were actually executed during training can be learned, and the focus on executing options in an on-policy manner, with option labels available. By relaxing these, we can improve sample efficiency and practical applicability, including the possibility to seed control policies from expert demonstrations.

**Contributions:** We present an algorithm that solves the problem of learning control abstractions by viewing the set of options as *latent variables* that *concisely represent* the agent's behaviour. More precisely, we do not only improve those options that were *actually* executed in a trajectory. Instead,

we allow intra-option learning by simultaneously improving all individual options that *could have been* executed, and the policy over options, in an end-to-end manner. We evaluate this algorithm on continuous MDP benchmark domains and compare it to earlier reinforcement learning methods that use flat and hierarchical policies.

## 2 RELATED WORK

Recent attention in the field of option discovery generally falls into one of two categories. One branch of work focuses on learning options that are able to reach specific subgoals within the environment. Much work in this category has focused on problems with discrete state and action spaces, indentifying salient or bottleneck states as subgoals (McGovern & Barto, 2001; Menache et al., 2002; Şimşek & Barto, 2009; Silver & Ciosek, 2012). Recent work has focused on finding subgoal states in continuous state spaces using clustering (Niekum & Barto, 2011) or spectral methods (Machado et al., 2017). Konidaris & Barto (2009) describes an approach where subgoals of new policies are defined by the initiation conditions of existing options. Specifying options using subgoals generally requires a given or a-priori learned system model, or specific assumptions about the environment. Furthermore, the policies to reach each subgoal have to be trained independently, which can be expensive in terms of data and training time (Bacon et al., 2017).

A second body of work has learned options by directly optimizing over the parameters of function approximations that are structured in a way to yield hierarchical policies. One possibility is to *augment* states or trajectories with the indexes of the chosen options. Option termination, selection, and inter-option behavior than all depend on both the regular system state and the current option index. This approach was suggested by Levy & Shimkin (2011) for learning the parameters of a hierarchical model consisting of pre-structured policies. In the option-critic architecture (Bacon et al., 2017), a similar model is employed, with option-specific value functions to learn more efficiently. Furthermore, neural networks are used instead of a task-specific given structure. Mankowitz et al. (2016) use an explicit partitioning of the state space to ensure policy specialization.

An alternative to state augmentation was proposed by Daniel et al. (2016). In that paper, options were considered latent variables rather than observable variables. That paper employed a policy structure that allowed maximizing the objective in the presence of these latent variables using an expectation-maximization approach. However, the optimization of this structure requires option policies to be linear in state features, which imposes the need to specify good state features a priori. Further, this approach necessitates the use of information from the entire trajectory before policy improvement can be done, eliminating the possibility of an on-line approach. Fox et al. (2017) uses a similar approach in the imitation learning setting with neural network policies instead of a task specific structure.

There are several other related works in hierarchical reinforcement learning outside of the options framework. One possibility is to have a higher-level policy learn to set goals for a learning lower-level policy (Vezhnevets et al., 2017), or to set a sequence of lower-level actions to be followed (Vezhnevets et al., 2016). Another possibility is to have a higher-level policy specify a prior over lower-level policies for different tasks, such that the system can acquire useful learning biases for new tasks (Wingate et al., 2011).

## 3 TECHNICAL BACKGROUND

We consider an agent interacting with its environment, at several discrete time steps. Generally, the state of the environment at step $t$, is provided in the form of a vector, $\mathbf{s}_t$, with $\mathbf{s}_0$ determined by an initial state distribution. At every step, the agent observes $\mathbf{s}_t$, and selects a vector-valued action $\mathbf{a}_t$, according to a stochastic policy $\pi(\mathbf{a}_t|\mathbf{s}_t)$, which gives the probability that an agent executes a particular action from a particular state. The agent then receives a reward $r_t$ and the next state $\mathbf{s}_{t+1}$ from the environment.

We consider episodic setups where, eventually, the agent reaches a terminal state, $\mathbf{s}_T$ upon which the environment is reset, to a state drawn from an initial state distribution. A sequence of states, actions and rewards generated in this manner is referred to as a trajectory $\tau$.

We define the discounted return from step $t$ within a trajectory to be $R_t^{(\tau)} = \sum_{i=t}^{T} \gamma^{(i-t)} r_i$. The objective of the learning agent is to maximize the expected per-trajectory return, given by $\rho = \mathbb{E}_\tau[R_0^{(\tau)}]$.

## 3.1 POLICY GRADIENT METHODS

While several methods exist for learning a policy from interaction with the environment, here, we focus on policy gradient methods, which have benefited from a recent resurgence in popularity. Policy gradient methods directly optimize $\rho$ by performing stochastic gradient ascent on the parameters $\boldsymbol{\theta}$ of a family of policies $\pi_{\boldsymbol{\theta}}$. Policy gradients can be estimated from sample trajectories, or in an online manner. The full return likelihood ratio gradient estimator (Williams, 1992) takes the form:

$$\nabla_{\boldsymbol{\theta}} \rho(\boldsymbol{\theta}) = \mathbb{E}_\tau \left[ (R_0^\tau - b) \sum_{t=0}^{T} \nabla_{\boldsymbol{\theta}} \log \pi(\mathbf{a}_t | \mathbf{s}_t) \right], \tag{1}$$

where $b$ is a baseline, used to reduce variance. This is one of the simplest, most general policy gradient estimators, and can be importance sampled if observed trajectories are not generated from the agent's policy. The policy gradient theorem (Sutton et al., 2000) expands on this result in the on-policy case, giving a gradient estimate of the form:

$$\nabla_{\boldsymbol{\theta}} \rho(\boldsymbol{\theta}) = \mathbb{E}_\tau \left[ \sum_{t=0}^{T} (R_t^\tau - b) \nabla_{\boldsymbol{\theta}} \log \pi(\mathbf{a}_t | \mathbf{s}_t) \right], \tag{2}$$

which can be shown to yield lower variance gradient estimates.

## 3.2 OPTIONS

The options framework provides the necessary formalism for abstraction over sequences of decisions in RL (Sutton et al., 1999; Precup, 2000). The agent is given access to a set of options, indexed by $\omega$. Each option has its own policy: $\pi_\omega(\mathbf{a}_t | \mathbf{s}_t)$, an initiation set, representing the states in which the option is available, and a termination function $\beta_\omega(\mathbf{s}_t)$, which represents the state-dependent probability of terminating the option. Additionally, the policy over options, $\pi_\Omega(\omega_t | \mathbf{s}_t)$ is employed to select from available options once termination of the previous option occurs.

During execution, option are used as follows: in the initial state, an option is sampled from the policy over options. An action is then taken according to the policy belonging to the currently active option. After selecting this action and observing the next state, the policy then terminates, or does not, according to the termination function. If the option does not terminate, the current option remains active. Otherwise the policy over options can be sampled in the new state in order to determine the next active option.

The policy over options can be combined with the termination function in order to yield the option-to-option policy function:

$$\tilde{\pi}_\Omega(\omega_t | \omega_{t-1}, \mathbf{s}_t) = [1 - \beta_{\omega_{t-1}}(\mathbf{s}_t)] \delta_{\omega_t \omega_{t-1}} + \beta_{\omega_{t-1}}(\mathbf{s}_t) \pi_\Omega(\omega_t | \mathbf{s}_t),$$

where $\delta$ is the Kronecker delta.

## 4 INFERRED OPTION POLICY GRADIENT

To learn options using a policy gradient method we parametrize all aspects of the policy: $\pi_{\Omega, \boldsymbol{\theta}}$ denotes the policy over options, parametrized by $\boldsymbol{\theta}$. $\pi_{\omega, \boldsymbol{\vartheta}}$ then denotes the intra-option policy of option $\omega$, parametrized by $\boldsymbol{\vartheta}$. Finally $\beta_{\omega, \boldsymbol{\xi}}$ is the termination function for $\omega$, parametrized by $\boldsymbol{\xi}$.

We aim to optimize the performance of the agent with respect to a set of policy parameters. The loss function is identical to that employed by traditional policy gradient methods: we optimize the expected return of trajectories in the MDP sampled using the current policy,

$$\rho(\boldsymbol{\theta}, \boldsymbol{\vartheta}, \boldsymbol{\xi}) = \mathbb{E}_\tau [R_\tau] = \int_\tau P(\tau) R_\tau \mathrm{d}\tau,$$

where $\mathbb{E}_\tau$ denotes expectation over sampled trajectories.

The expected performance can be maximized by increased the probability of visiting highly rewarded state-action pairs. To increase this probability, it does not matter which option originally generated that state-action pair, rather, we will derive an algorithm that updates all options that *could have* generated that state-action pair. Determining these options is done in a differentiable inference step. As a result the policy can be optimized end-to-end, yielding our *Inferred Option Policy Gradient* algorithm.

In order to compute the gradient of the loss objective, we decompose $P(\tau)$ into the relevant conditional probabilities, and employ the "likelihood ratio" method, so that it is possible to estimate the gradient from samples:

$$\nabla\rho = \int_\tau P(\tau)R_\tau \left[ \sum_{i=0}^{T} \nabla \log P(\mathbf{a}_i|\mathbf{s}_{[0:i]}, \mathbf{a}_{[0:i-1]}) \right] \mathrm{d}\tau.$$

Note that this is similar to the REINFORCE policy gradient, though here actions are not independent, even when conditioned on states, since information can still pass through the unobserved options.

In order to compute the inner gradient, we marginalize over the hidden options at each time step, leading to:

$$\nabla\rho = \int_\tau P(\tau)R_\tau \left[ \sum_{i=0}^{T} \nabla \log \left( \sum_{\omega_i} P(\omega_i|\mathbf{s}_{[0:i]}, \mathbf{a}_{[0:i-1]})\pi_{\omega_i,\vartheta}(\mathbf{a}_i|\mathbf{s}_i) \right) \right] \mathrm{d}\tau.$$

Figure 1: Graphical Model for Option Trajectory

Recognizing the hidden Markov model-like structure of the trajectories shown in Fig. 1 reveals that the $P(\omega_i|\mathbf{s}_{[0:i]}, \mathbf{a}_{[0:i-1]})$ term can be expressed in a recursive form, simply as an application of the *forward* algorithm:

$$P(\omega_i|\mathbf{s}_{[0:i]}, \mathbf{a}_{[0:i-1]}) = \sum_{\omega_{i-1}} c_i^{-1} P(\omega_{i-1}|\mathbf{s}_{[0:i-1]}, \mathbf{a}_{[0:i-2]})\pi_{\omega_{i-1},\vartheta}(\mathbf{a}_{i-1}|\mathbf{s}_{i-1})\tilde{\pi}_{\Omega,\theta,\xi}(\omega_i|\omega_{i-1}, \mathbf{s}_i)$$

where $c_i$ is a normalization factor, given by:

$$c_i = \sum_{\omega_{i-1}} P(\omega_{i-1}|\mathbf{s}_{[0:i-1]}, \mathbf{a}_{[0:i-2]})\pi_{\omega_{i-1},\vartheta}(\mathbf{a}_{i-1}|\mathbf{s}_{i-1}).$$

and our initial value is $P(\omega_0|\mathbf{s}_0) = \pi_{\Omega,\theta}(\omega_0|\mathbf{s}_0)$. If our policies are differentiable, then this recursive term is differentiable as well, allowing us to perform gradient descent to maximize our objective, using the sampled data to compute the full return Monte Carlo gradient estimate:

$$\nabla\rho \approx R_0 \left[ \sum_{i=0}^{T} \nabla \log \left( \sum_{\omega_i} P(\omega_i|\mathbf{s}_{[0:i]}, \mathbf{a}_{[0:i-1]})\pi_{\omega_i,\vartheta}(\mathbf{a}_i|\mathbf{s}_i) \right) \right],$$

where $\tau = (\mathbf{s}_0, \mathbf{a}_0, \ldots, \mathbf{a}_{T-1}, \mathbf{s}_T)$ is a trajectory sampled from the system using the current policy $\pi_\theta$. The variance of this estimator can be reduced through inclusion of a constant baseline, through an argument identical to that used for REINFORCE (Williams, 1992).

Here, we notice that actions at any given time step are conditionally independent of rewards received in the past, given the trajectory up that action. As in other policy gradient methods, we can reduce variance further by removing these terms from our gradient estimator. This is formally expressed as:

$$\forall j < k \quad \mathbb{E}_{\mathbf{s}_{[0:k]}, \mathbf{a}_{[0:k]}} \left[ r_j \nabla \log P(\mathbf{a}_k | \mathbf{s}_{[0:k]}, \mathbf{a}_{[0:k-1]}) \right] = 0.$$

With this realization, we can simplify our estimator to:

$$\nabla \rho \approx \left[ \sum_{i=0}^{T} \left( \sum_{j=i}^{T} (r_j) - b(\mathbf{s}_j) \right) \nabla \log \left( \sum_{\omega_i} P(\omega_i | \mathbf{s}_{[0:i]}, \mathbf{a}_{[0:i-1]}) \pi_{\omega_i, \boldsymbol{\vartheta}}(\mathbf{a}_i | \mathbf{s}_i) \right) \right], \quad (3)$$

where $b(\mathbf{s}_j)$ is a state-dependent baseline. Note that the estimate is unbiased regardless of the baseline, although good baselines can reduce the variance. In this work, we use a learned parametric approximation of the value function $V_{\boldsymbol{\nu}}$ as baseline. The value function is learned using gradient descent on the mean squared prediction error of Monte-Carlo returns:

$$\nabla_{\boldsymbol{\nu}} \sum_{t=0}^{T} (V_{\boldsymbol{\nu}}(\mathbf{s}_t) - R_t)^2. \quad (4)$$

Estimating the value function can also be done using other standard methods such as LSTD or TD($\lambda$).

Below, we describe the algorithm for learning options to optimize returns through a series of interactions with the environment. While Algorithm 1 can only be applied in the episodic RL setup, it is also possible to employ the technical insight shown here in an online manner. One potential method for doing so is described in Appendix A.

---

**Algorithm 1:** Inferred Option Policy Gradient

---

Initialize parameters randomly
**foreach** *episode* **do**
    $\omega_0 \sim \pi_\Omega(\omega | \mathbf{s}_0)$             // sample an option from the policy over options at the initial state
    **for** $t \leftarrow 0, \dots, T$ **do**
        $\mathbf{a}_t \sim \pi_{\omega_t}(\mathbf{s}_t)$             // sample an action according to the current intra-option policy
        Get next state $\mathbf{s}_{t+1}$ and reward $r_t$ from the system
        $\omega_{t+1} \sim \tilde{\pi}_\Omega(\omega_{t+1} | \omega_t, \mathbf{s}_{t+1})$      // sample the next option according to the policy over option
    **end**
    Update $\boldsymbol{\nu}$ according to (4), using sampled episode
    $\boldsymbol{\theta}$, $\boldsymbol{\vartheta}$, and $\boldsymbol{\xi}$ according to (3), using sampled episode
**end**

---

## 5 EXPERIMENTS

In order to evaluate the effectiveness of our algorithm, as well as the qualitative attributes of the options learned, we examine its performance across several standardized continuous control environments as implemented in the OpenAI Gym (Brockman et al., 2016) in the MuJoCo physics simulator (Todorov et al., 2012). In particular, we examine the Hopper-v1 (observation dimension: 11, action dimension: 3), Walker2d-v1 (observation dimension: 17, action dimension: 6), HalfCheetah-v1 (observation dimension: 17, action dimension: 6), and Swimmer-v1 (observation dimension: 8, action dimension: 2) environments. Generally, they all require the agent to learn to operate joint motors in order to move the agent in a particular direction, with penalties for unnecessary actions. Together, they are considered to be reasonable benchmarks for state-of-the art continuous RL algorithms.

### 5.1 COMPARISON OF PERFORMANCE

We compared the performance of our algorithm (IOPG) with results from option-critic (OC) and asynchronous actor-critic (A3C) methods, as described in Mnih et al. (2016).

In order to ensure an appropriate comparison, IOPG and OC were also implemented using multiple agents operating in parallel, as is done in A3C. The option-critic algorithm as described in Bacon

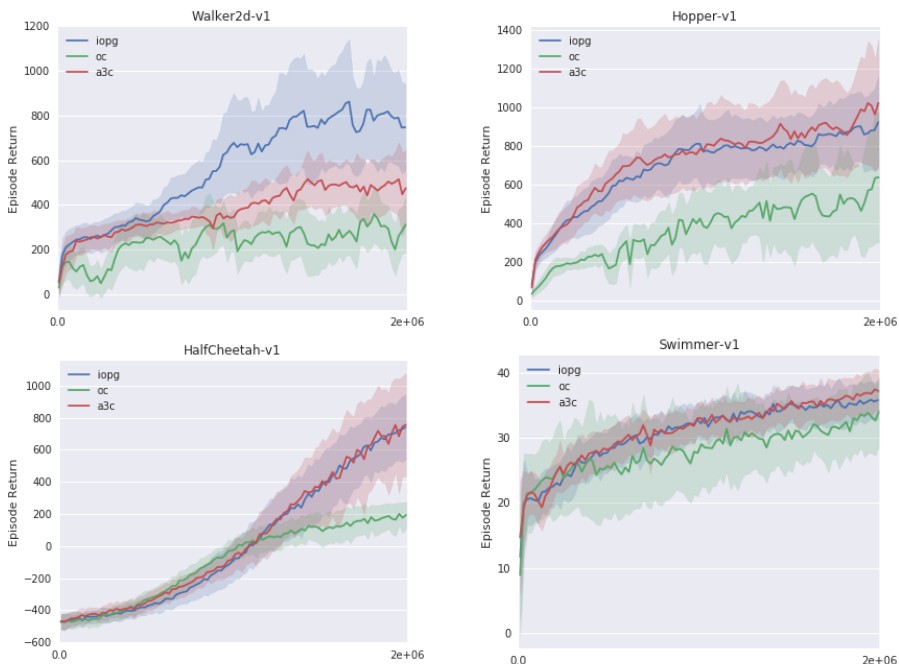

Figure 2: Training curves for 2 million time steps averaged across 10 random seeds for several continuous RL domains. The shaded area represents the 95% confidence interval.

et al. (2017) employs greedy option selection according to the learned Q. To ensure a fair comparison, we employed the same parametrized actor as the inter-option policy in our option-critic baseline as was used in IOPG. Since option-critic already learns option-value functions, no SMDP-level value function approximation is needed.

Our model architecture for all three algorithms closely follows that of Schulman et al. (2017). The policies and value functions were represented using separate feed-forward neural networks, with no parameters shared. For each agent, both the value function and the policies used two hidden layers of 64 units with $\tanh$ activation functions. The IOPG and OC methods shared these parameters across all policy and termination networks. The option sub-policies and A3C policies were implemented as linear layers on top of this, representing the mean of a Gaussian distribution. The variance of the policy was parametrized by a linear softplus layer. Option termination was given by a linear sigmoid layer for each option. The policy over options, for OC and IOPG methods, was represented using a final linear softmax layer, of size equal to the number of options available. The value function for IOPG and AC methods was represented using a final linear layer of size 1, and for OC, size $|\Omega|$. All weight matrices were initialized to have normalized rows.

RMSProp (Tieleman & Hinton, 2012) was used to optimize parameters for all agents. We employ a single shared set of RMSProp parameters across all asynchronous threads. Additionally, entropy regularization was used during optimization for the AC policies, the option policies and the policies over options. This was done in order to encourage exploration, and to prevent the policies from converging to single repeated actions, as policy gradient methods parametrized by neural networks often suffer from this problem.

The results of these experiments are shown in Fig. 2. We see that IOPG, despite having significantly more parameters to optimize, and recovering additional structure, is able to learn as quickly as A3C across all of the domains, and learns significantly faster in the Walker2d environment. This is likely enabled by the fact that all of the options in IOPG can make use of all of the data gathered. OC, on the other hand seems to suffer a reduction in learning speed due to the fact that options are not all learned simultaneously, preventing experience from being shared between them.

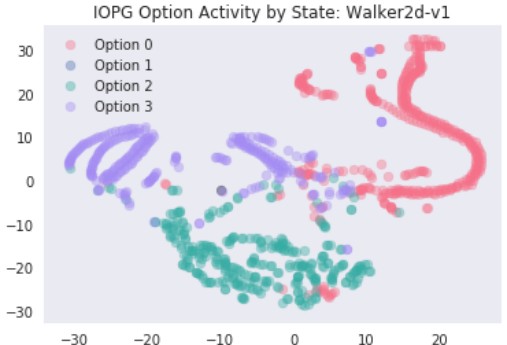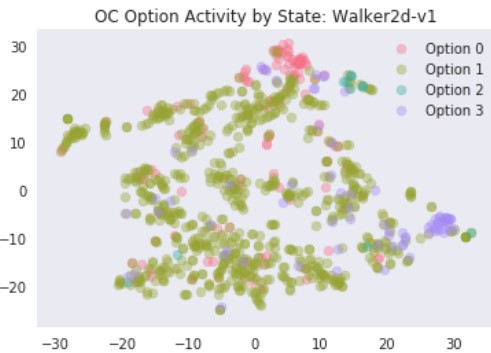

Figure 3: Typical option activity as a function of state. The axes represent T-SNE embeddings of higher dimensional states. Each state is coloured according to the option that was active at that point in the trajectory. We can see that the options learned by Option-Critic (Right) are not visibly correlated in state-space, while options learned by IOPG (Left) are.

## 5.2 OPTION STRUCTURE

In order to further understand the nature of the options learned, we performed a visualization of them over a random subsample of states in the last 8000 frames. We perform T-SNE (Maaten & Hinton, 2008) on these states in order to represent the high-dimensional state space in two dimensions, while preserving some structure.

Fig. 3 shows the results of this procedure. We can see that different options are active in different regions of state space. This indicates that the options learned can be interpreted as having some local structure. Options appear to be spatially coherent, as well as having structure in the policy. The relation between state and action abstraction has been observed previously in the RL literature (Andre & Russell, 2002; Provost et al., 2007). It is also likely that options employed are temporally coherent, since in smooth, continuous domains, it is likely the case that spatially close states are also close in time, matching the intuitive notion that options represent abstract behaviours, which can extend over several actions.

Fig. 4 displays additional analyses of the options learned in the Walker2d environment. We found that in this particular environment, agents with either four or eight options available perform roughly equally, while having only two options led to sub-optimal performance (Fig. 4a). This effect can be explained by the fact that three options seem to be sufficient, and if more options are given only three of them tend to get frequently selected (Fig. 4b). This finding suggests that only three of the options that IOPG learns are useful here, perhaps due to the relative simplicity of the environment. In Fig. 4c, we observe further evidence that the options learned by IOPG are temporally extended. A moving average of the continuation probability $(1 - \beta_\omega(s_t))$ during training indicates that early on, when the options are not well optimized, termination occurs quite frequently. As the options improve, termination decreases, until the policy over options is only queried approximately every ten steps on average.

## 6 DISCUSSION

In this paper, we have introduced a new algorithm for learning hierarchical policies within the options framework, called inferred option policy gradients. This algorithm treats options as latent variables. Gradients are propagated through a differentiable inference step that allows end-to-end learning of option policies, as well as option selection and termination probabilities.

In our algorithms policies take responsibility for state-actions pairs they *could have* generated. In contrast, in learning algorithms for hierarchical policies that use an augmented state space, option policies are updated using only those state-action pairs the actually generated. As a result, in our algorithm options do not tend to become 'responsible' for unlikely states or actions they generated. Thus, options are stimulated more strongly to specialize in a part of the state space. We conjecture that this specialization caused the discussed increase in the interpretability of options.

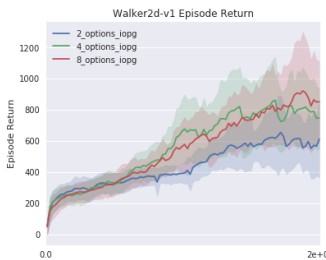

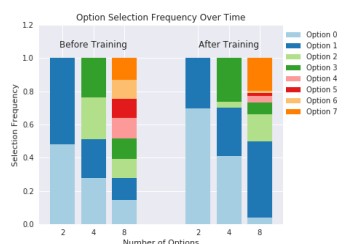

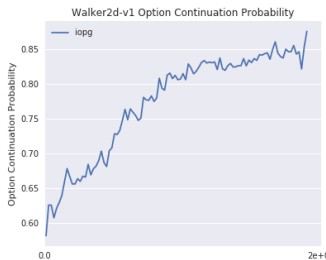

(a) Performance in the Walker2d environment as a function of available options. We see that in this environment, having several options available to the agent leads to an improved policy.

(b) In the Walker2d environment, initially option selection is uniform. After training only 3 options tend to be selected, even when more are available. Selection frequencies are averaged over 100 sampled states.

(c) As the options improve during training, the probability of remaining in the active option increases, plateauing at around 0.85. This suggests that the options learned here exhibit temporal extension.

Figure 4: Analysis of the learned options in the Walker2d environment.

Furthermore, in our experiments learning with inferred options was significantly faster than learning with an option-augmented state space. In fact, learning with inferred options proved equally fast, or sometimes even faster, than using a comparable non-hierarchical policy gradient method despite IOPG having many more parameters. We conjecture that option inference encourages intra-option learning, thus allowing multiple options to improve as the result of a single learning experience, causing this speed-up.

In future work, we want to quantify the suitability of the learned options for transfer between tasks. Our experiments so far were in the episodic setting. We want to investigate an on-line, actor-critic version of learning with inferred options to learn continuously in infinite-horizon problems.

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

## A APPENDIX: ONLINE GRADIENT ESTIMATES

In addition to the batch gradient estimator described in Section 4, it is also possible to develop an online estimator. While we introduce some bias by updating the parameters in the middle of the trajectory, this increases the number of environments in which our method can be applied. In order to achieve this, we employ a method similar to real-time recurrent learning (Williams & Zipser, 1989), leveraging the recursive structure of the gradient estimate in order to make computation tractable.

For convenience, let $\eta(t)$ denote the vector of option probabilities at time step $t$. That is to say $\eta(t)_\omega = P(\omega_t|\mathbf{s}_{[0:t]}, \mathbf{a}_{[0:t-1]})$. Also, let $\boldsymbol{\psi}$ denote the concatenation of parameter vectors $\boldsymbol{\theta}, \boldsymbol{\vartheta}$, and $\boldsymbol{\xi}$. Through a simple application of the chain rule, we observe that:

$$\frac{\partial P(\omega_t|\mathbf{s}_{[0:t]}, \mathbf{a}_{[0:t-1]})}{\partial \boldsymbol{\psi}} = \frac{\partial P(\omega_t|\mathbf{s}_{[0:t]}, \mathbf{a}_{[0:t-1]})}{\partial \eta} \frac{\partial \eta(t-1)}{\partial \boldsymbol{\psi}} + \frac{\partial \eta(t)_\omega}{\partial \boldsymbol{\psi}} \tag{5}$$

Thus, in order to efficiently compute this gradient in an online manner, in addition to our parameter vector $\boldsymbol{\psi}$, we maintain an additional set of gradient traces, $g_{\omega\psi}$, for each option, and update them according to equation 5. These values are then substituted for $\frac{\partial \eta(t-1)}{\partial \boldsymbol{\psi}}$ when computing the subsequent gradient. This procedure adds an additional memory complexity of $O(|\boldsymbol{\psi}| \times |\Omega|)$, since a gradient trace over all parameters must be maintained for each option.

An inferred option actor-critic (IOAC) algorithm using this gradient estimator is described below. Note that this algorithm–in addition to learning online–could exhibit lower variance than the IOPG method described above. By using a learned estimator for the returns instead of the Monte Carlo results, updates are more consistent, ideally leading to increased stability, at the cost of some bias.

---

**Algorithm 2:** Inferred Option Actor Critic

initialize $\boldsymbol{\psi}$ randomly
**for** *e in episodes* **do**
    $g_{\omega\psi} \leftarrow \mathbf{0}$
    $\mathbf{s} \leftarrow \mathbf{s}_0$
    $\omega \sim \pi_\Omega(\mathbf{s})$
    **for** *t in timesteps* **do**
        $\mathbf{a} \sim \pi_\omega(\mathbf{a}|\mathbf{s})$
        $\mathbf{s}', r \sim \texttt{step}(\mathbf{a}, \mathbf{s})$
        Update $\boldsymbol{\nu}$ according to TD
        Update $g_{\omega\psi}$ according to (5)
        Substitute $g_{\omega\psi}$ into (3) to update $\boldsymbol{\theta}$ and $\boldsymbol{\vartheta}$
        Draw option termination $b \sim \beta(\mathbf{s}')_\omega$
        **if** $b$ **then**
            $\omega \sim \pi_\Omega(\mathbf{s}')$
        **end**
        $\mathbf{s} \leftarrow \mathbf{s}'$
    **end**
**end**

---

