# OpenReview forum: "An inference-based policy gradient method for learning options"
_ICLR.cc/2018/Conference — Reject_

### Official Review · AnonReviewer2 · 2017-11-27
**This paper proposes a new algorithm for discovering options, but the benefits of the algorithm are not clear empirically.**

**Rating:** 3
**Confidence:** 4

**Review:**

This paper treats option discovery as being analogous to discovering useful latent variables.  The proposed formulation assumes there is a policy over options, which invokes an option’s policy to select actions at each timestep until the option’s termination function is activated.  A contribution of this paper is to learn all possible options that might have caused an observed trajectory, and to update parameters for all these pertinent option-policies with backprop.  The proposed method, IOPG, is compared to A3C and the option-critic (OC) on four continuous control tasks in Mujoco, and IOPG has the best performance on one of the four domains.

The primary weakness of this paper is the absence of performance or conceptual improvements in exchange for the additional complexity of using options.  The only domain where IOPG outperforms both A3C and OC is the Walker2D-v1 domain, and the reported performance on that domain (~800) is far below the performance of other methods (shown on OpenAI’s Gym site or in the PPO paper). Also, there is not much analysis on what kind of options are learned with this approach, beyond noting that the options seem clustered on tSNE plots.  Given the close match between the A3C agent and the IOPG agent on the other three domains, I expect that the system is mostly relying on the base A3C components with limited contributions from the extensions introduced in the network for options.

The clarity of the paper’s contributions could be improved. The contribution of options might be made more clearly in smaller domains or in more detailed experiments. How is the termination beta provided from the network?  How frequently did the policy over options switch between them?  How was the number of options selected, and what happens when the number of possible options is varied from 1 to 4 or beyond 4?  To what extent was there overlap in the learned policies to realize the proposed algorithmic benefit of learning multiple option-policies from the same transitions?  The results in this paper do not provide strong support for using the proposed method.

---

> ### Author Response · Authors · 2017-12-14
> **Response**
>
> Thank you to the reviewer for their insightful comments. Individual points are addressed below.
>
>
> > The only domain where IOPG outperforms both A3C and OC is the Walker2D-v1 domain, and the reported performance on that domain (~800) is far below the performance of other methods (shown on OpenAI’s Gym site or in the PPO paper).
>
> We do not expect IOPG to outperform A3C significantly in a single task, but expect benefits in interpretability and transferability. Please see our comment for our more detailed response to this. We would like to add here that we view the option learning strategy contributed here to be largely independent to the method used for policy optimization. This is to say that it should be easy to write a PPO-IOPG algorithm with the benefits of both. We compare to A3C so that the value of our contribution in isolation is more clear. We could also add a comparison to an inferred-option extension to PPO.
>
>
> > How is the termination beta provided from the network?
>
> We apologize for forgetting to include this. Termination is sampled for the currently active option from a linear sigmoid layer on top of the policy network, as an additional head. We will clarify this in the updated version of the paper.
>
>
> > How frequently did the policy over options switch between them?
>
> We will add this information to the appendix in the next version of the paper.
>
>
> > How was the number of options selected, and what happens when the number of possible options is varied from 1 to 4 or beyond 4?
>
> Most existing option learning methods require specification of the number of options as a hyperparameter. In general this is optimized according to the task at hand. Here, however, we did no optimization over this parameter, but we'll be happy to add an experiment to the next version of the paper.
>
>
> > To what extent was there overlap in the learned policies to realize the proposed algorithmic benefit of learning multiple option-policies from the same transitions?
>
> At the start of learning, the policies tend to overlap highly due to random initialization. Because of this, early training benefits from the simultaneous update, as all options are implicated in every action. As training progresses, the t-SNE experiments demonstrate that is little overlap between final policies. Each policy appears to be active in a different region of state space. This is likely due to the fact that the most likely option is most updated, rather than a single option being updated improperly in the event of an unlikely action.

---

### Official Review · AnonReviewer1 · 2017-11-29
**The paper is well written and presents a good extension of infernece based option discovery. However, the results are not convincing and there is a crucial issue in the assumptions of the algorithm.**

**Rating:** 3
**Confidence:** 5

**Review:**

The paper presents a new policy gradient technique for learning options. The option index is treated as latent variable and, in order to compute the policy gradient, the option distribution for the current sample is computed by using a forward pass. Hence, a single sample can be used to update all options and not just the option that has been used for this sample.

The idea of the paper is good but the novelty is limited. As noted by the authors, the idea of using inference for option discovery has already been presented in Daniel2016. Note that the option discovery process is Daniel2016 is not limited to linear sub-policies, only the policy update strategy is. So the main contribution is to use a new policy update strategy, i.e., policy gradients, for inference based option discovery. Thats fine but should be stated more clearly in the paper. The paper is also written very well and the topic is relevant for the ICLR conference.

However, the paper has two main problems:
- The results are not convincing. In most domains, the performance is similar to the A3C algorithm (which does not use inference based option discovery), so the impact of this paper seems limited.

- One of the main assumptions of the algorithm is wrong. The assumption is that rewards from the past are not correlated with actions in the future conditioned on the state s_t (otherwise we would always have a correlation) ,which  is needed to use the policy gradient theorem. The assumption is only true for MDPs. However, using the option index as latent variable yields a PoMDP. There, this assumption does not hold any more. Example: Reward at time step t-1 depends on the action, which again depends on the option o_t-1. Action at time step t depends on o_t. Hence, there is a strong correlation between reward r_t-1 and action a_t+1 as o_t and o_t+1 are strongly correlated. o_t is not a conditional variable of the policy as it is not part of the state, thats why this assumption does not work any more.

Summary: The paper is well written and presents a good extension of inference based option discovery. However, the results are not convincing and there is a crucial issue in the assumptions of the algorithm.

---

> ### Author Response · Authors · 2017-12-14
> **Response**
>
> We thank the reviewer for taking the time to evaluate our paper. Individual points are addressed below.
>
>
> > The assumption [that rewards are independent of future actions, conditioned on the current state] is only true for MDPs. However, using the option index as latent variable yields a PoMDP. There, this assumption does not hold any more.
>
> Under the standard set of assumptions this would be correct. As shown in the line before Eqn. 3, the conditional assumption that we make is slightly different. It is true that a_k and r_j are not independent in general. However, they are conditionally independent given s_k and s_j, and a_j. We are conditioning on all of the observed states and observed actions since the start of the trajectory. Since the reward only depends on these observed variables, no information is passed to future actions.
>
>
> > As noted by the authors, the idea of using inference for option discovery has already been presented in Daniel2016. Note that the option discovery process is Daniel2016 is not limited to linear sub-policies, only the policy update strategy is. So the main contribution is to use a new policy update strategy, i.e., policy gradients, for inference based option discovery
>
> We agree that the graphical model employed here is the same as that used in Daniel2016. However, the option inference step is not the same, since they employ the use of backward information, while we only require forwards information. This means that our algorithm can be employed online, while the one presented in Daniel2016 can only be applied in the episodic case, where updates are made only after the episode is terminated.
>
>
> > The results are not convincing. In most domains, the performance is similar to the A3C algorithm (which does not use inference based option discovery), so the impact of this paper seems limited.
>
> We do not expect IOPG to outperform A3C significantly in a single task, but expect benefits in interpretability and transferability. Please see our official comment for our more detailed response to this.

---

### Official Review · AnonReviewer3 · 2017-12-05
**Interesting, but not impactful**

**Rating:** 4
**Confidence:** 4

**Review:**

This paper proposes what is essentially an off-policy method for learning options in complex continuous problems.  The idea is to use policy gradient style algorithms to update a suite of options using relatively

On the positive side, I like the core idea of this paper.  The idea of updating multiple options at once is a good one.  I think the authors should definitely continue to investigate this line of work.  I also appreciated that the authors took the time to try and visualize what was learned.  The paper is generally well-written and easy to read.

On the negative side: ultimately, the algorithm doesn't seem to work all that well.  Empirically, the method doesn't seem to perform substantially better than other algorithms, although there seems to be some slight advantage.  A clearly missing comparison would be something like TRPO or DDPG.

Figure 1 was helpful in understanding marginalization and the forward algorithm.  Thanks.

Was there really only 4 options that were learned?  How would this scale to more?

---

> ### Author Response · Authors · 2017-12-14
> **Response**
>
> We thank the reviewer for their time and insight. Individual points are addressed below.
>
>
> > Empirically, the method doesn't seem to perform substantially better than other algorithms, although there seems to be some slight advantage.  A clearly missing comparison would be something like TRPO or DDPG.
>
> We do not expect IOPG to outperform A3C significantly in a single task, but expect benefits in interpretability and transferability. Please see our official comment for our more detailed response to this. While TRPO has been shown to outperform A3C in certain situations, we feel that the policy update strategy is largely independent of the option learning method presented here. That is, it should not be too difficult to write an algorithm that uses trust region updates with option learning. We compare to A3C so that the value of our contribution in isolation is more clear. We could also add a comparison to an inferred-option extension of a more powerful policy search algorithm such as PPO, TRPO, or DDPG.
>
>
> > Was there really only 4 options that were learned?  How would this scale to more?
>
> The number of options learned is prespecified as a hyperparameter, as is the case in several option learning methods. The computational complexity is quadratic in the number of options, with linear memory complexity. We will add an experiment comparing the number of options in the next version of the paper.

---

### Author Response · Authors · 2017-12-14
**Performance of IOPG**

We would like to thank the reviewers for their insightful comments. Here, we focus on the issue that all three reviewers raised: that A3C does as well as IOPG in most environments.

It is our opinion that A3C ought to perform roughly as well as IOPG. The optimization performed is nearly identical between the two algorithms, where IOPG is parameterized in a particular manner such that options can be learned. We developed IOPG as a data-efficient method to optimize several options simultaneously. We present it in a general form, without any sort of regularization on the structure of the options. Even without such regularization, the options learned by IOPG express some worthwhile characteristics, which several existing option learning algorithms cannot produce: namely temporal extension, and spatial separation. Without additional problem-specific regularization on the structure of those options, there is no reason to expect performance improvements in the single-task setting.

This said, we feel that the extra structure learned by IOPG yields several benefits. Options can be useful for the interpretation of agent behaviours, as our t-SNE experiments (Fig. 3) show. Further there is strong evidence to suggest that learned options can be useful for transfer learning (OptionGAN: Henderson et al. 2018, Option-Critic: Bacon et al. 2017, Subgoal Discovery: McGovern and Barto 2001). We feel that these benefits make IOPG a worthwhile algorithm, especially since it comes at no cost to data efficiency, variance, or asymptotic learning compared to A3C. We are currently working on experiments that better quantify such upsides.

---

### Author Response · Authors · 2018-01-05
**Updated Version**

We have uploaded an updated version of the paper, which addresses some of the concerns the reviewers had, as well as providing additional information on the nature of the options learned.

---

### Decision · Program_Chairs · 2018-01-29
**ICLR 2018 Conference Acceptance Decision**

**Decision:**

Reject

**Comment:**

The reviewers are unanimous that this is an interesting paper, but that ultimately the empirical results are not sufficiently promising to warrant the added complexity.